# Multilayer-HySEA model validation for landslide generated tsunamis. Part I Rigid slides

Jorge Macías*, Cipriano Escalante, Manuel J. Castro

*Departamento de Análisis Matemático, Estadística e Investigación Operativa y Matemática Aplicada, Facultad de Ciencias, Universidad de Málaga, 29080-Málaga*

## Abstract

This paper is devoted to benchmarking the Multilayer-HySEA model using laboratory experimental data for landslide-generated tsunamis. This article deals with rigid slides and the second part, in a companion paper, addresses granular slides. The US National Tsunami Hazard and Mitigation Program (NTHMP) has proposed the experimental data used and established for the NTHMP Landslide Benchmark Workshop, held in January 2017 at Galveston (Texas). The first three benchmark problems proposed in this workshop deal with rigid slides. Rigid slides must be simulated as a moving bottom topography and, therefore, they must be modelled as a prescribed boundary condition. These three benchmarks are used here to validate the Multilayer-HySEA model. This new HySEA model consists of an efficient hybrid finite-volume/finite-difference implementation on GPU architectures of a non-hydrostatic multilayer model. A brief description of model equations, dispersive properties, and the numerical scheme is included. The benchmarks are described and the numerical results compared against the lab-measured data for each of them. The specific aim is to validate this new code for tsunamis generated by rigid slides. Nevertheless, the overall objective of the current benchmarking effort is to produce a ready-to-use numerical tool for real-world landslide generated tsunami hazard assessment. This tool has already been used to reproduce the Port Valdez Alaska 1964 and Stromboli Italy 2002 events.

*Keywords:* Multilayer-HySEA model, tsunamis, rigid slides, model benchmarking, landslide-generated tsunamis, GPU implementation

*Corresponding author
Email address:* `jmacias@uma.es` (Jorge Macías)

*Preprint submitted to Natural Hazards and Earth Systems Science*                    *December 31, 2020*

*2010 MSC:* 35L, 65-05, 76-05, 86-08

## 1. Introduction

Model development and benchmarking for earthquake-induced tsunamis is a task addressed in the past and to which much effort and time has been dedicated. In particular, just to mention a couple of NTHMP efforts, the 2011 Galveston benchmarking workshop (Horrillo et al., 2015) and the 2015 Portland workshop for tsunami currents (Lynett et al., 2017) both focused on these topics. However, both model development and benchmarking efforts have advanced at a slower pace for landslide-generated tsunamis. As examples we might mention the 2003 NSF-sponsored landslide tsunami workshop organized in Hawaii and a similar follow-up workshop on Catalina Island in 2006 (Liu et al., 2008). Since then, no similar large comprehensive benchmarking workshop has been organized (Kirby et al., 2018).

In its 2019 Strategic Plan, the NTHMP required that all numerical tsunami inundation models to be used in hazard assessment studies in the US be verified as accurate and consistent through a model benchmarking process. This mandate was fulfilled in 2011, but only for seismic tsunami sources and to a limited extent for idealized solid underwater landslides. However, recent work by various NTHMP states has shown that landslide tsunami hazard may in fact be greater than seismically-induced hazard and may be also the dominant risk along significant parts of the US coastline (ten Brink et al., 2014).

As a result of this demonstrated gap, a set of candidate benchmarks was proposed to perform the required validation process. The selected benchmarks are based on a subset of available laboratory data sets for solid slide experiments and deformable slide experiments and include both submarine and subaerial slides. In order to complete this list of laboratory data, a benchmark based on a historic field event (Valdez, AK, 1964) was also chosen. The EDANYA group (www.uma.es/edanya) from the University of Málaga participated in the workshop organized at Texas A&M University, Galveston (January 9-11,

2017) presenting results for the benchmarking tests with two numerical codes: Landslide-HySEA and Multilayer-HySEA models. At Galveston, we presented numerical results for six out of the seven benchmark problems proposed, including the field case. The current work presents the numerical results obtained for the Multilayer-HySEA model in the framework of the validation effort described above for the case of rigid slide-generated tsunamis, whereas the benchmark problems dealing with granular slides are presented in a companion paper Macías et al. (2020a). A summary of the results for the field case at Port Valdez can be found at Macías et al. (2017).

Twenty years ago, at the beginning of the century, the challenge of solid block landslide modelling was taken by a number of researchers (Grilli and Watts, 1999, 2005; Grilli et al., 2002; Lynett and Liu, 2002; Watts et al., 2003; Wu, 2004; Watts et al., 2005; Liu et al., 2005) and laboratory experiments were developed for those cases and for tsunami model benchmarking (Enet and Grilli, 2007) (see also Ataie-Ashtiani and Najafi-Jilani (2008)). The benchmark problems performed in the current work are based on the laboratory experiments of Grilli and Watts (2005) for BP1, Enet and Grilli (2007) for BP2, and Wu (2004); Liu et al. (2005) for BP3. The basic reference for these three benchmarks, as well as for the three benchmarks related to granular slides and the Alaska field case (all of them proposed by the NTHMP) is Kirby et al. (2018). We highly recommend checking this reference for further details on benchmark descriptions, data provided for performing them, required benchmark items, and inter-model comparison. Finally, we would like to stress that the ultimate goal of our current benchmarking effort is to provide the tsunami community with a NTHMP-approved model for landslide-generated tsunami hazard assessment, similarly to what we have done with the Tsunami-HySEA model for the case of earthquake-generated tsunamis (Macías et al., 2017; Macías et al., 2020c,d).

## 2. HySEA models for landslide generated tsunamis

The HySEA (Hyperbolic Systems and Efficient Algorithms) software consists of a family of geophysical codes based on either single-layer, two-layer stratified systems or multilayer shallow-water models. HySEA codes[1] have been under development by the EDANYA Group from UMA (the University of Málaga) for more than a decade. These codes are in continuous evolution and upgrading and they are serving to a wider scientific community every day. The first model we developed dealing with landslide-generated tsunamis consisted of a stratified two-layer Savage-Hutter shallow-water model -the Landslide-HySEA model-. It was implemented based on the model described in Fernández et al. (2008) and was incorporated to the HySEA family. An initial validation of this code, comparing numerical results with the laboratory experiments of Heller and Hager (2011) and Fritz et al. (2001) can be found at Sánchez-Linares (2011). The 2018 numerical simulation of the Lituya Bay 1958 mega-tsunami with real topo-bathymetric data and encouraging results (González-Vida et al., 2019), represented a milestone in the verification process of this code. This validation effort was accomplished under a research contract with PMEL/NOAA. The result of this project led the NCTR (NOAA Center for Tsunami Research) to adopt Landslide-HySEA as the numerical code of choice to generate the initial conditions for the MOST model to be initialized in the case of a landslide-generated tsunami scenario to be simulated. Further applications of Landslide-HySEA can be found at de la Asunción et al. (2013), Macías et al. (2015), and Iglesias (2015).

The waves generated in the laboratory tests proposed in the NTHMP selected benchmarks are high frequency and dispersive, and the generated flows have a complex vertical structure. Therefore, the numerical model used must be able to reproduce such effects. This makes the two-layer Landslide-HySEA model unsuitable for reproducing these experimental results as non-hydrostatic

---

[1]https://edanya.uma.es/hysea

effects play an important role and a richer vertical structure is required. To address these requirements, the Multilayer-HySEA model was very recently implemented, considering a stratified structure in the simulated fluid and including non-hydrostatic terms. A multilayer model is able to better approximate the vertical structure of a complex flow than a standard one-layer depth-averaged model. In particular, increasing the number of layers the linear dispersion relation of the model converges towards the exact dispersion relation from the Stokes linear theory (see Fernández-Nieto et al. (2018)).

## 3. Model Equations

The Multilayer-HySEA model implements one of the multilayer non-hydrostatic models of the family introduced and described in Fernández-Nieto et al. (2018) (model $LDNH_0$). The governing equations, that are obtained by a process of depth-averaging, correspond to a semi-discretization for the vertical variable of the Euler equations following a standard Galerkin approach. The total pressure is decomposed into a sum of hydrostatic and non-hydrostatic pressures and is assumed to have a linear vertical profile. The horizontal and vertical velocities are assumed to have a constant vertical profile. At the discrete level on z, the total pressure matches at the interfases and velocities satisfy a discrete jump condition (see Fernández et al. (2008) or Escalante et al. (2018a)).

An alternative deduction for this system is performed in Escalante et al. (2018a) assuming linear vertical profiles for pressure and vertical velocity and a constant vertical profile for the horizontal velocity, as well as some extra hypothesis for the case of two layers. The proposed model admits an exact energy balance and, when the number of layers increases, the linear dispersion relation of the linear model converges to the same of Airy's theory (Fernández-Nieto et al., 2018). The model proposed in Fernández-Nieto et al. (2018) can

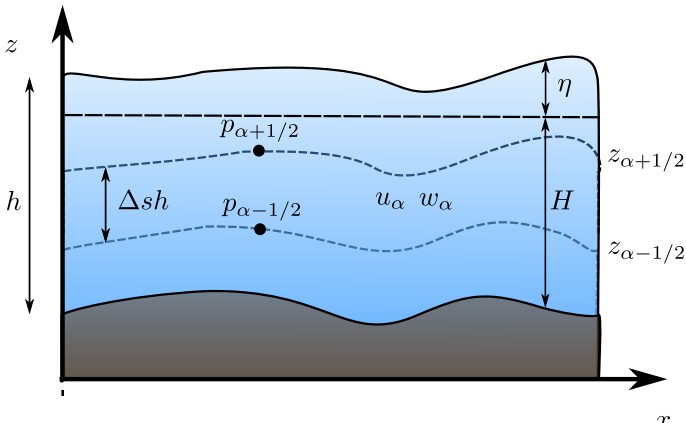

Figure 1: Schematic diagram describing the multilayer system

be written in a compact form as:

$$
\begin{cases}
\partial_t h + \partial_x \left( h\bar{u} \right) = 0, \\[2ex]
\partial_t \left( hu_\alpha \right) + \partial_x \left( hu_\alpha^2 + \dfrac{1}{2} gh^2 \right) - gh\partial_x H + u_{\alpha+1/2}\Gamma_{\alpha+1/2} - u_{\alpha-1/2}\Gamma_{\alpha-1/2} = \\[1ex]
\hspace{6cm} - h \left( \partial_x p_\alpha + \sigma_\alpha \partial_z p_\alpha \right) - \tau, \\[2ex]
\partial_t \left( hw_\alpha \right) + \partial_x \left( hu_\alpha w_\alpha \right) + w_{\alpha+1/2}\Gamma_{\alpha+1/2} - w_{\alpha-1/2}\Gamma_{\alpha-1/2} = -h\partial_z p_\alpha, \\[2ex]
\partial_x u_{\alpha-1/2} + \sigma_{\alpha-1/2}\partial_z u_{\alpha-1/2} + \partial_z w_{\alpha-1/2} = 0,
\end{cases}
\tag{1}
$$

where, for $\alpha \in \{1,\ 2,\ \ldots, L\}$, the following notation is used:

$$
f_{\alpha+1/2} = \frac{1}{2} \left( f_{\alpha+1} + f_\alpha \right), \ \partial_z f_{\alpha+1/2} = \frac{1}{h\Delta s} \left( f_{\alpha+1} - f_\alpha \right),
\tag{2}
$$

where $f$ denotes one of the generic variables of the system, i.e., $u,\ w$ and $p$, and, finally,

$$
\sigma_\alpha = \partial_x \left( H - h\Delta s(\alpha - 1/2) \right), \ \sigma_{\alpha-1/2} = \partial_x \left( H - h\Delta s(\alpha - 1) \right).
\tag{3}
$$

Total depth, $h$, is split along the vertical axis into $L \geq 1$ layers and $\Delta s = 1/L$ (see Figure 1). The variables $u_\alpha$ and $w_\alpha$ are the depth-averaged velocities in

the $x$ and $z$ directions, respectively, $t$ is time and $g$ is gravitational acceleration. The non-hydrostatic pressure at the interface $z_{\alpha+1/2}$ is denoted by $p_{\alpha+1/2}$. The water surface elevation measured from the still-water level is $\eta = h - H$, where $H$ is the water depth when the water is at rest. Finally, $\tau$ is a friction law term, and the terms $\Gamma_{\alpha+1/2}$ account for the mass transfer across interfaces and are defined by

$$\Gamma_{\alpha+1/2} = \sum_{\beta=\alpha+1}^{L} \partial_x \left( h \Delta s \left( u_\beta - \bar{u} \right) \right), \ \bar{u} = \sum_{\alpha=1}^{L} \Delta s u_\alpha \tag{4}$$

In order to close the system of equations, the following boundary conditions are considered

$$p_{L+1/2} = 0, \ u_0 = 0, \ w_0 = -\partial_t H. \tag{5}$$

Note that the motion of the bottom surface can be taken into account as a boundary condition, imposing $w_0 \neq 0$. Therefore, this model can simulate the interaction with a slide in the case that the motion of the bottom is prescribed by a function, given by a set of data, or simulated by a numerical model. In the present study, we are going to consider tests where the motion of the seafloor is given by a known function (the solid moving block).

## 3.1. Linear dispersion relation

Some dispersive properties of the system (1) are presented in this subsection, in particular, the phase and group velocities, and the linear shoaling. The first two properties are related to the propagation of dispersive wave trains and the last one to shoaling processes.

To obtain such properties, the system (1) is linearised around the water at rest steady-state solution. After that, a Stokes-type Fourier analysis is carried out looking for first-order planar wave solutions. This method constitutes a standard procedure to study systems that model dispersive water waves (see Escalante et al. (2018a); Lynett and Liu (2004); Madsen and Sorensen (1992); Schäffer and Madsen (1995) and references therein). The phase and group

velocities as well as the linear shoaling gradient are, respectively, defined as:

$$C = \omega/k, \quad G = C + k\partial_k C, \quad \frac{\partial_x \eta}{\eta} = -\gamma \frac{\partial_x H}{H}, \tag{6}$$

where $\omega$ denotes the angular frequency, $k$ the local wave-number and $H$ the typical depth.

The measured quantities $C$, $G$ and $\gamma$ are solely functions of the local wave-number and the typical depth $H$. Thus, one can obtain the so-called linear dispersion relation of the three measured quantities. From the Airy wave theory, one can also obtain the corresponding linear dispersion relations that state the linear theory for the considered quantities (see Schäffer and Madsen (1995) for the Airy reference formulae). For example, the expression for the phase velocity from the Airy's theory is

$$C_{Airy} = gH \frac{\tanh(kH)}{kH}. \tag{7}$$

The expressions of the phase velocity for the system (1) are given in Table 1 for the non-linear hydrostatic shallow water system (SWE) and the Multilayer-HySEA (non-hydrostatic) system with $j \geq 1$ layers (NH–jL). The last two columns contain $Er_C(s)$ for $s = 5$ and $s = 15$, where $Er_C(s)$ represents the maximum relative error of the phase velocity with respect to the Airy in a range $kH \in [0, s]$ in percent, i.e.:

$$Er_C(s) = 100 \cdot \max_{kH \in [0,s]} \left( \frac{|C(kH) - C(kH)_{Airy}|}{|C(kH)_{Airy}|} \right). \tag{8}$$

The main goal when deriving dispersive shallow water systems is to get the most accurate dispersive relations as possible, compared with the Airy wave theory, without highly increasing the complexity of the system. See Schäffer and Madsen (1995) for a review on state-of-the-art or a two-layer with improved dispersive relations in Lynett and Liu (2004), and an enhanced two-layer non-hydrostatic pressure system in Escalante et al. (2018a). It has been shown (Fernández-Nieto et al., 2018) that increasing the number of layers leads to the convergence of the linear dispersion relation of the linear model to the same of Airy's theory. Figure 2 shows this behavior and highlights the huge discrepancies

| Multilayer System – Phase velocity – Errors for $kH$ up to 5 and 15 | | | |
|---|---|---|---|
| Model | Phase velocity | $Er_C(5)$ | $Er_C(15)$ |
| (SWE) | $gH$ | 73.63 % | 123.61 % |
| (NH-1L) | $gH\dfrac{1}{1+\frac{1}{4}(kH)^2}$ | 3.02 % | 16.95 % |
| (NH-2L) | $gH\dfrac{1+\frac{(kH)^2}{16}}{1+\frac{3(kH)^2}{8}+\frac{(kH)^4}{256}}$ | 0.71 % | 10.67 % |
| (NH-3L) | $\dfrac{1+\frac{5(kH)^2}{54}+\frac{(kH)^4}{1296}}{1+\frac{5(kH)^2}{12}+\frac{5(kH)^4}{432}+\frac{1(kH)^6}{46656}}$ | 0.31 % | 0.62 % |
| (NH-5L) | $\dfrac{1+\frac{3(kH)^2}{25}+\frac{63(kH)^4}{2510^3}+\frac{3(kH)^6}{2510^4}+\frac{(kH)^8}{1010^7}}{1+\frac{9(kH)^2}{20}+\frac{21(kH)^4}{1010^2}+\frac{21(kH)^6}{1010^4}+\frac{9(kH)^8}{2010^6}+\frac{(kH)^{10}}{1010^9}}$ | 0.11 % | 0.11 % |

Table 1: Phase velocity expressions and maximum of the relative error $Er_C(s)$ compared with the Airy's theory for different ranges of $kH \in [0, s]$ for the non-linear hydrostatic shallow water system (SWE) and the Multilayer-HySEA (non-hydrostatic) system with $j \geq 1$ layers (NH–jL).

between the Airy's theory and the systems (SWE) and (NH-1L). It is well known that waves generated by landslides, might present high characteristic values for $kH$. For the (SWE) system, it is well known that it has an accurate phase velocity in a small range of $kH$, and that this system is appropriate for long waves as tsunami waves, but not for dispersive waves with higher values of $kH$. In the same vein, the one layer non-hydrostatic pressure system (NH-1L) can improve these results, but again, poor linear dispersive results are achieved in

 a range of $kH$ between 5 and 15. However, when the number of layers, $L$, is set

to 3 (still a small value) the system (1) is in an excellent agreement with the

Airy theory for $kH$ up to 15. For the phase celerity, the percentage error is less

than 0.62%, and for the group velocity is less than 1% for $kH$ smaller than 10

(see Figure 2). Linear shoaling is also well reproduced in this same range.

The Multilayer-HySEA model presents enhanced dispersive properties. In

order to have similar dispersive results as the ones obtained here using a three-

layer system, at least five layers are required for other similar multilayer models

as the one presented in Bai and Cheung (2018). Furthermore, the results pre-

sented for the phase velocity with two layers in Table 1 show that the system

proposed here produces smaller relative error for $kH$ up to 15 compared with the

two-layer system in Cui et al. (2014). That means that the Multilayer-HySEA

model can achieve better dispersive properties than models having similar or

even more computational complexity.

### 3.2. Modeling of breaking waves and wetting and drying treatment

### 3.2.1. Modeling of breaking waves

In shallow areas the breaking of waves can be observed near the coast. As

pointed out in Escalante et al. (2018a,b, 2019); Roeber et al. (2010) among

others, non-hydrostatic PDE systems such as the one considered in this paper

cannot describe this process without the inclusion of an additional term that

accounts for the dissipation of the amount of energy required when breaking

phenomena occur. In this work, we have implemented a simplified generalization

of the breaking mechanism that was introduced in Escalante et al. (2018a) for

the case of two layers. To do so, the vertical component of the stress-tensor is

depth-averaged on the vertical variable. Thus, adding the proposed integrated

viscosity term to system (1), only the vertical momentum equation changes and

reads for each $\alpha \in \{1, 2, \ldots, L\}$ as:

$$\partial_t \left( h w_\alpha \right) + \partial_x \left( h u_\alpha w_\alpha \right) + w_{\alpha+1/2} \Gamma_{\alpha+1/2} - w_{\alpha-1/2} \Gamma_{\alpha-1/2} = -h \partial_z p_\alpha + 2 \varsigma w_\alpha, \quad (9)$$

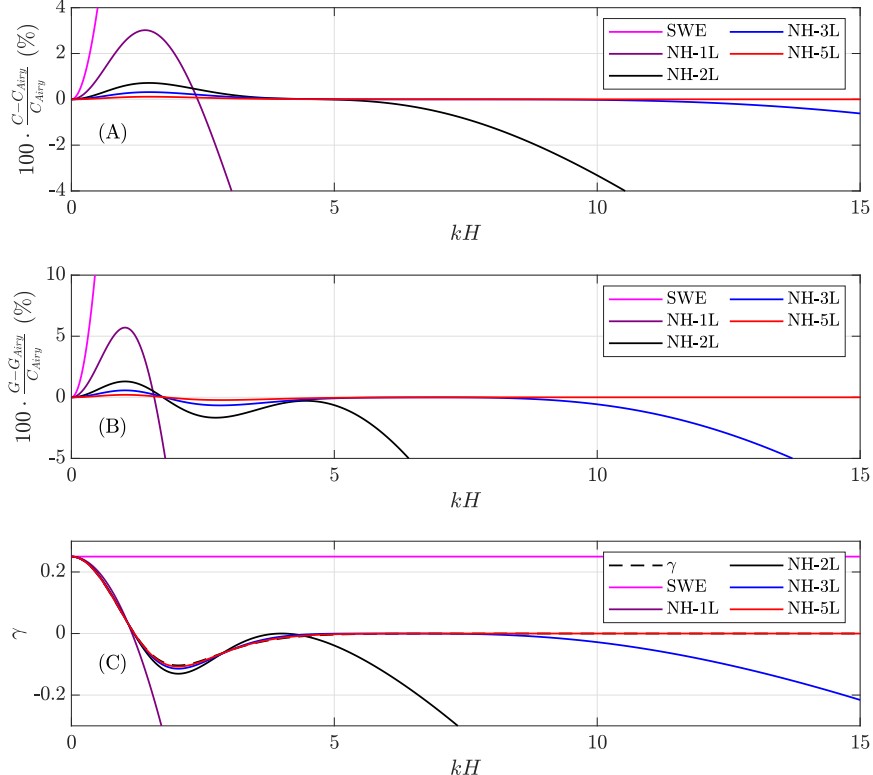

Figure 2: Relative error for the phase velocities (A), the group velocities (B), and comparison with the reference shoaling gradient (C), with respect to the Airy theory for the described multilayer systems (2L, 3L, and 5L), the one-layer non-hydrostatic, and the shallow-water system.

where $\varsigma = \int_{z_{\alpha-1/2}}^{z_{\alpha-1/2}} \partial_z \nu$ is the eddy viscosity. In this work, as in Escalante et al. (2018a); Roeber et al. (2010) we choose $\varsigma$ to be

$$\varsigma = -KBh \left| \partial_x(h\bar{u}) \right|, \tag{10}$$

where $B$ is an empirical parameter related to a breaking criteria to switch on and off this extra dissipation term. The definition of this empirical parameter is based on a quasi-heuristic strategy to determine when the breaking occurs (see Escalante et al. (2018b); Roeber et al. (2010) and references therein).

Finally, a natural and simple extension of the criterion proposed by Roeber

et al. (2010) is adopted

$$B = \max\left(1 - \frac{\partial_x(h\bar{u})}{U_1},\ 0\right) \quad \text{for} \quad |\partial_x(h\bar{u})| \geq U_2, \tag{11}$$

where

$$U_1 = B_1\sqrt{gh}, \quad U_2 = B_2\sqrt{gh} \tag{12}$$

denote the flow speed at the onset and termination of the wave-breaking process and $B_1$, $B_2$ are calibration coefficients. In this work, we use $B_1 = 0.6$ and $B_2 = 0.15$ for all the test cases studied. Finally, depending on the benchmark problem, we use $K \in \{2, 10\}$.

### 3.2.2. Wetting and drying treatment

For the computation of variables in areas of small water depth, a wet-dry treatment adapting the ideas described in Castro et al. (2005) is applied. The key elements for the numerical treatment of wet-dry fronts with emerging bottom topographies are based on:

- The hydrostatic pressure terms $\partial_x\left(\frac{1}{2}gh^2\right) - gh\partial_x H = gh\partial_x\eta$ at the horizontal velocity equations are modified for emerging bottoms to avoid spurious pressure forces (see Castro et al. (2005)).

- To overcome the difficulties due to large round-off errors in computing the velocities $u_\alpha, w_\alpha$ from discharges for small values of $h$, we define the velocities analogously as in Kurganov and Petrova (2007) applying the desingularization formula

$$u_\alpha = \frac{\sqrt{2}hhu_\alpha}{\sqrt{h^4 + \max(h^4, \epsilon^4)}}, \ w_\alpha = \frac{\sqrt{2}hhw_\alpha}{\sqrt{h^4 + \max(h^4, \epsilon^4)}}, \ \alpha \in \{1, 2, \dots, L\} \tag{13}$$

  which gives the exact value of $u_\alpha$ and $w_\alpha$ for $h \geq \epsilon$, and gives a smooth transition of $u_\alpha$ and $w_\alpha$ to zero when $h$ tends to zero without truncation. In this work we set $\epsilon = 10^{-3}$ for the numerical tests. A more detailed discussion about the desingularization formula can be seen in Kurganov and Petrova (2007).

## 4. Numerical Solution Method

We describe now the discretization of system (1) that follows the natural extension of the procedure described in Escalante et al. (2018a,b) for the one and two layer non-hydrostatic system. The numerical scheme employed is based on a two-step projection-correction method, similar to the standard Chorin's projection method for Navier-Stokes equations (Chorin (1968)). This is a standard procedure when dealing with dispersive systems (see Escalante et al. (2018b,a); Ma et al. (2012); Kazolea and Delis (2013); Ricchiuto and Filippini (2014) and references therein).

First, we shall solve the non-conservative hyperbolic underlying system (1) given by the compact equation

$$\partial_t \mathbf{U} + \partial_x \mathbf{F}_{SW}(\mathbf{U}) + \mathbf{B}_{SW}(\mathbf{U})\partial_x \mathbf{U} = \mathbf{G}_{SW}(\mathbf{U})\partial_x H, \tag{14}$$

where the following compact notation has been used:

$$\mathbf{U} = \begin{pmatrix} h \\ hu_1 \\ \vdots \\ hu_L \\ hw_1 \\ \vdots \\ hw_L \end{pmatrix}, \ \mathbf{F}_{SW}(\mathbf{U}) = \begin{pmatrix} hu \\ \dfrac{hu_1^2}{h} + \dfrac{1}{2}gh^2 \\ \vdots \\ \dfrac{hu_L^2}{h} + \dfrac{1}{2}gh^2 \\ hu_1 w_1 \\ \vdots \\ hu_L w_L \end{pmatrix}, \ \mathbf{G}_{SW}(\mathbf{U}) = \begin{pmatrix} 0 \\ gh \\ \vdots \\ gh \\ 0 \\ \vdots \\ 0 \end{pmatrix}, \tag{15}$$

and $\mathbf{B}_{SW}$ is a matrix such $\mathbf{B}_{SW}\partial_x \mathbf{U}$ contains the non-conservative products related to the mass transfer across interfaces appearing at the momentum equations.

Then, in a second step, non-hydrostatic terms given by the pressure vector

correction term

$$
\mathcal{T}_{\mathcal{NH}}(h, \partial_x h, H, \partial_x H, \mathbf{P}, \partial_x \mathbf{P}) = - \begin{pmatrix} 0 \\ h\left(\partial_x p_1 + \sigma_1 \partial_z p_1\right) \\ \vdots \\ h\left(\partial_x p_L + \sigma_L \partial_z p_L\right) \\ h\partial_z p_1 \\ \vdots \\ h\partial_z p_L \end{pmatrix}, \quad \mathbf{P} = \begin{pmatrix} p_1 \\ p_2 \\ \vdots \\ p_L \end{pmatrix},
$$

(16)

as well as the divergence constraints at each layer

$$
\mathcal{B}(\mathbf{U}, \partial_x \mathbf{U}, H, \partial_x H) = \begin{pmatrix} \partial_x u_{1/2} + \sigma_{1/2} \partial_z u_{1/2} + \partial_z w_{1/2} \\ \vdots \\ \partial_x u_{L-1/2} + \sigma_{L-1/2} \partial_z u_{L-1/2} + \partial_z w_{L-1/2} \end{pmatrix}, \quad (17)
$$

will be taken into account.

System (14) is discretized using a second order finite volume PVM positive-preserving well-balanced path-conservative method (Castro and Fernández-Nieto, 2012). As usual, we consider a set of $N$ finite volume cells $I_i = [x_{i-1/2}, x_{i+1/2}]$ with constant lengths $\Delta x$ and define

$$
\mathbf{U}_i(t) = \frac{1}{\Delta x} \int_{I_i} \mathbf{U}(x, t) \, dx, \tag{18}
$$

the cell average of the function $\mathbf{U}(x, t)$ on cell $I_i$ at time $t$. Concerning non-hydrostatic terms, we consider mid-points $x_i$ of each cell $I_i$ and denote the point values of the function $\mathbf{P}$ at time $t$ by

$$
\mathbf{P}_i(t) = \begin{pmatrix} p_1(x_i, t) \\ p_2(x_i, t) \\ \vdots \\ p_L(x_i, t). \end{pmatrix} \tag{19}
$$

Non-hydrostatic terms will be approximated by second order compact finite-differences.

Let us detail de time stepping procedure followed. Assume given time steps $\Delta t^n$, and denote $t^n = \sum_{k \leq n} \Delta t^k$. To obtain second order accuracy in time, the two-stage second-order TVD Runge-Kutta scheme is adopted. At the $k$th stage, $k \in \{1, 2\}$, the two-step projection-correction method is given by

$$
\begin{cases}
\dfrac{\mathbf{U}^{(\widetilde{k})} - \mathbf{U}^{(k-1)}}{\Delta t} + \partial_x \mathbf{F}(\mathbf{U}^{(k-1)}) + \mathbf{B}(\mathbf{U}^{(k-1)})\partial_x \mathbf{U}^{(k-1)} & \text{(20a)} \\
\qquad\qquad\qquad\qquad\qquad = \mathbf{G}(\mathbf{U}^{(k-1)})\partial_x H, & \\[2mm]
\dfrac{\mathbf{U}^{(k)} - \mathbf{U}^{(\widetilde{k})}}{\Delta t} = \mathcal{T}(h^{(k)}, \partial_x h^{(k)}, H, \partial_x H, \mathbf{P}^{(k)}, \partial_x \mathbf{P}^{(k)}) & \text{(20b)} \\[4mm]
\mathcal{B}(\mathbf{U}^{(k)}, \partial_x \mathbf{U}^{(k)}, H, \partial_x H) = \mathbf{0} & \text{(20c)}
\end{cases}
$$

where $\mathbf{U}^{(0)}$ is $\mathbf{U}$ at the time level $t^n$, $\mathbf{U}^{(\widetilde{k})}$ is an intermediate value in the two-step projection-correction method that contains the numerical solution of the hyperbolic system (14) at the corresponding $k$th stage of the Runge-Kutta, and $\mathbf{U}^{(k)}$ is the $k$th stage estimate. After that, a final value of the solution at the $t^{n+1}$ time level is obtained:

$$
\mathbf{U}^{n+1} = \frac{1}{2}\mathbf{U}^n + \frac{1}{2}\mathbf{U}^{(2)}. \tag{21}
$$

Observe that, equations (20b-20c) requires, at each stage of the calculation respectively, to solve a Poisson-like equation for each one of the variables contained in $\mathbf{P}^{(k)}$. The resulting linear system is solved using an iterative Jacobi method combined with a scheduled relaxation (see Adsuara et al. (2016); Escalante et al. (2018a,b)). Note that the usual CFL restriction must be imposed for the computation of the time step $\Delta t$.

With respect to the breaking mechanism introduced in Subsection 3.2, these terms are semi-implicitly discretized at the end of the second step of the proposed numerical scheme, at each Runge-Kutta stage. The resulting numerical scheme is well-balanced for the water at rest solution and is linearly $L^\infty$-stable under the usual CFL condition related to the hydrostatic system. It is also worth mentioning that the numerical scheme is positive preserving and can deal with emerging topographies. Finally, its extension to 2D is straightforward. In

this case, the computational domain is decomposed into subsets with a simple geometry, called cells or finite volumes. The numerical algorithm is well suited for its implementation in GPU architectures, as is shown in Castro et al. (2011). Furthermore, the compactness of the numerical stencil and the natural and the massively parallelization of the Jacobi method makes it possible that the second step can also be implemented in GPUs (see Escalante et al. (2018b,a)). That results in a high efficiency of the numerical code and much shorter computational times.

## 5. Benchmark Problem Comparisons

In this Section, the numerical results obtained with the Multilayer-HySEA model and the comparison with the measured lab data for waves generated by the movement of a rigid bottom surface or of a solid block are presented. In particular, BP1 deals with a 2D submarine solid slide, BP2 with a 3D submarine slide and, finally, BP3 consists of two 3D slides, one partially submerged and a second one representing a completely submarine slide. In all these cases, a moving bottom condition has been used to model the solid block movement. Regarding the wave breaking model, the breaking mechanism described in Subsection 3.2 was implemented, adopting $B_1 = 0.6$, $B_2 = 0.15$ for all the benchmark problems, and $K = 10$ for the third benchmark and $K = 2$ for the rest. The description of all these benchmarks can be found at LTMBW (2017) and Kirby et al. (2018).

### 5.1. Benchmark Problem 1: Two-dimensional submarine solid block

This benchmark problem is based on the 2D laboratory experiments of Grilli and Watts (2005) which were performed at the University of Rhode Island. Refer to the above-mentioned work to get a detailed description of the present benchmark. Figure 3 depicts the sketch of the laboratory experiment design. The 2D slide model is semi-elliptical, lead-loaded, and rolling down a smooth slope with a slope angle $\theta = 15°$ (2 mm above the slope), in between two vertical

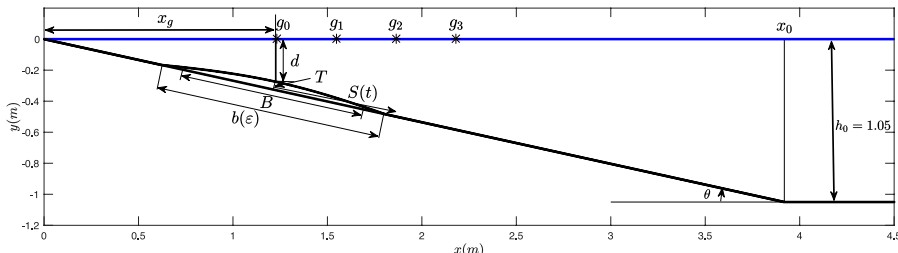

Figure 3: BP1. Sketch of main parameters and variables for wave generation by 2D rigid slide. [Modified from Grilli and Watts, 2005].

side walls, 20 cm apart. The water depth is $h_0 = 1.05$ m over the flat bottom part. The slide dimensions were, length B $= 1$ m, maximum thickness $T = T_{\mathrm{ref}}$ $= 0.052$ m, and width $w = 0.2$ m. The model initial submergence $d$ was varied in experiments and the free surface elevation recorded at 4 capacitance wave gauges installed at locations: $x' = 1.175$, $1.475$, $1.775$, and $2.075$ m, the first location being nearly identical to $x'_g = 1.168$ m (where de tilde variables, as $x'$, mean than non-dimensional units are used -see Table 3-).

| $x'_g$ | $T'$ | $d'$ | $\theta$ | $B$ | $b(\epsilon)$ |
|---|---|---|---|---|---|
| 1.168 | 0.052 | 0.259 | 15 | 1 | 1.225 |

Table 2: Values for variables defining setup configuration.

| | $g_0$ | $g_1$ | $g_2$ | $g_3$ |
|---|---|---|---|---|
| $x$ | 1.234 | 1.549 | 1.864 | 2.179 |
| $x' = x/h_0$ | 1.175 | 1.475 | 1.775 | 2.075 |

Table 3: Gauge positions in dimensional and non-dimensional units.

In this benchmark, two items remained not completely determined in the original description provided: the first one is related with the initialization of the numerical experiment, the second one is related with how and where the solid moving block must stop. Other small issues related to the description of the benchmark were put forward in Macías et al. (2017) at our NTHMP report.

The motion of the rigid slide was prescribed as a function of time as

$$S(t) = S_0 \, \log(\cosh(t/t_0)), \qquad (22)$$

where $S_0 = u_t^2/a_0 = 2.110 \, \text{m}$, $t_0 = u_t/a_0 = 1.677 \, \text{s}$, $a_0 = 0.75 \, \text{m/s}^2$ and $u_t = 1.258 \, \text{m/s}$ is the terminal velocity. Figure 4 shows the prescribed acceleration, velocity and rigid slide displacement. In the laboratory experiment, the block is stopped at time $t = t_0 = 1.667 \, s$. and we replicate this behaviour in the numerical model. We also performed numerical experiments (not presented here) where the block continued moving at constant speed.

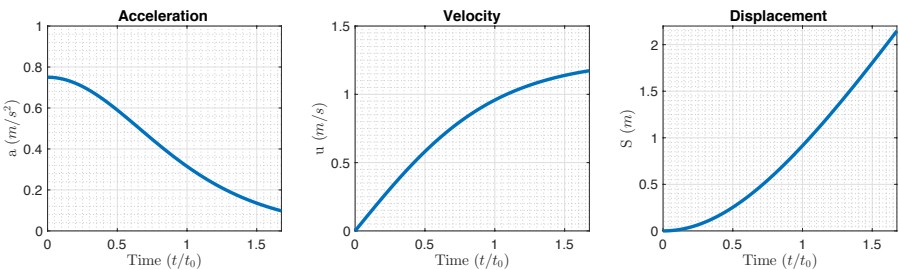

Figure 4: BP1. Prescribed acceleration, velocity and displacement of the solid slide.

The benchmark here consists of using the above information on slide shape, submergence, and kinematics, together with reproducing the experimental set-up to simulate surface elevations measured at the four wave gauges (average of 2 replicates of experiments provided).

Then, in order to reproduce the lab experiment, the interval $[-1, 10]$ discretized with $\Delta x = 0.02 \, m$, is the computational domain considered. In the vertical, taking three layers seems to produce optimal results. Increasing the number of layers gives similar results increasing the computational cost. The stability $CFL$ number was set to 0.9 and $g = 9.81$. The numerical simulation performed was 4 $s$ long in real time. As boundary conditions, outflow conditions were imposed at $x = -1$, $x = 10$.

In Figure 5 the comparison of the numerical results with the filtered lab measured data is presented. A good overall agreement between them can be observed. Some discrepancies can be seen after draw-down in all the gauges.

This behavior could also be observed, except for the last gauge, at Grilli and
Watts (2005) results. These authors explained that this behavior could be due
to unwanted surface tension effects. Given this comparison, and considering
the experimental variations and errors inherent to laboratory work and data
processing, it can be concluded that the Multilayer-HySEA model performs
optimally the present benchmark test.

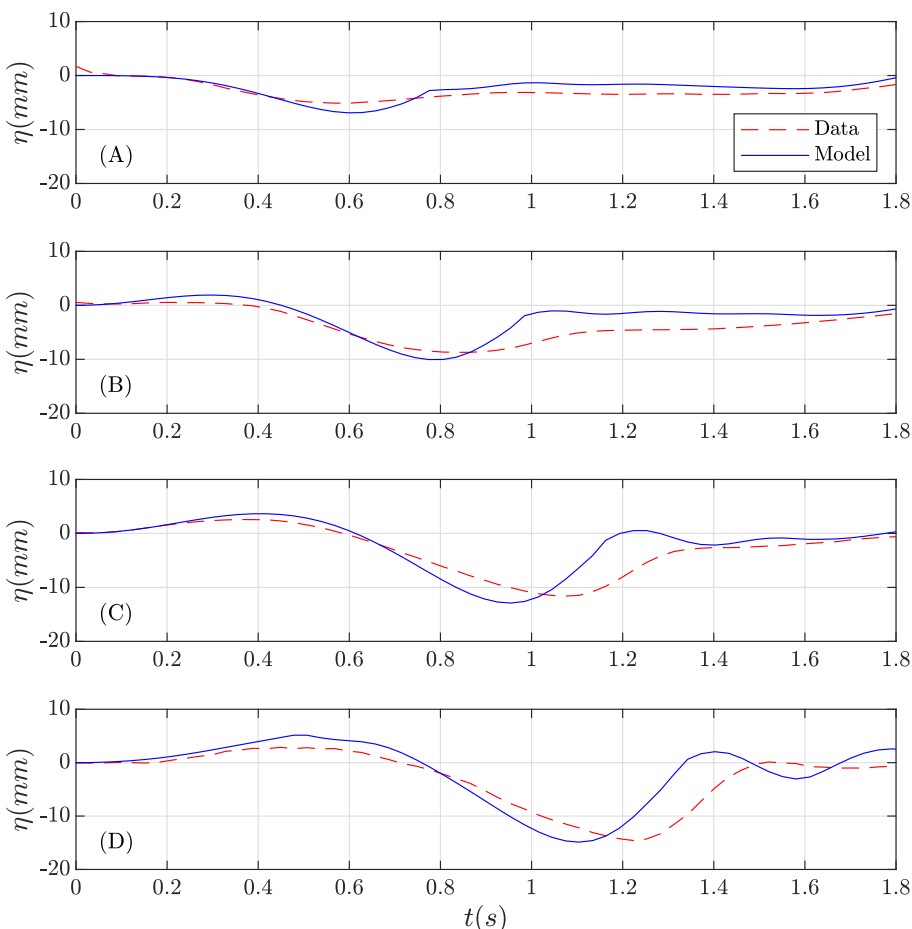

Figure 5: BP1. Filtered data (in red) and numerically simulated (in blue) time series at wave
gauges (A) $g_0$, (B) $g_1$, (C) $g_2$, and (D) $g_3$.

*5.2. Benchmark Problem 2: Three-dimensional submarine solid block*

This second benchmark consists of a 3D extension of BP1. The longitudinal sketch of the experiment is the same as in Figure 3. In the horizontal plane, cross-sections are elliptic, the plan view of the rigid slide, for the case $d = 61\,mm$, is presented in Figure 6. It is based on the 3D laboratory experiments of Enet and Grilli (2007). The experiments were also performed at the University of Rhode Island in a water wave tank of width 3.6 m and length 30 m, with a still water depth of 1.5 m over the flat bottom portion. As in the previous benchmark, the angle of the plane slope where the slide slid down is $\theta = 15°$. The submarine slide model was built as a streamline Gaussian-shaped aluminum body with elliptical footprint (see Figure 6), with down-slope length $b = 0.395$ m, cross-slope width $w = 0.680$ m, and maximum thickness $T = 0.082$ m. Complete details about the analytic definition of the slide shape and the experimental setting can be found at Kirby et al. (2018) and at LTMBW (2017).

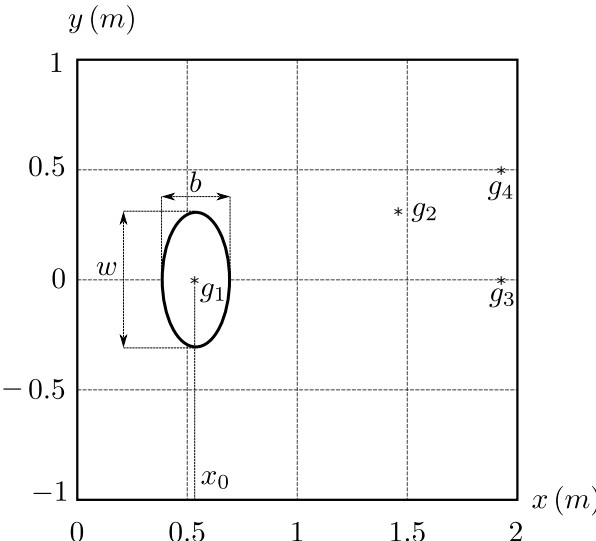

Figure 6: BP2. Sketch of the plan view (case $61\,mm$). [From Kirby et al. (2018)].

For the numerical simulations, the two-dimensional computational domain $[-1, 10] \times [-1.8, 1.8]$ is considered and discretized with $\Delta x = \Delta y = 0.02\,m$. The number of layers was set to 3. Numerical tests were performed using more layers

and similar results were obtained. The $CFL$ number was set to 0.9 and $g = 9.81$.
The simulated time was $6\,s$. As boundary conditions, rigid wall conditions were
imposed at $y = -1.8,\ y = 1.8$ and outflow conditions at $x = -1,\ x = 10$.

The benchmark test proposed consists in reproducing the slide shape and
complete experimental set-up in and using the information about submergence
and kinematics to replicate numerically Enet and Grilli's experiments for $d =$
61 and $d = 120$ mm. It is required to simulate surface elevations measured
at the four wave gauges (average of 2 replicates of experiments) and present
comparisons of the model with the experimental results.

Enet and Grilli (2007) performed experiments for 7 initial submergence
depths $d$. They are listed in Table 4, together with values of related slide
parameters and some measured tsunami wave characteristics. Here, the numer-
ical results corresponding to the two NTHMP required experiments (for $d = 61$
and $d = 120$ mm) will be presented first, then, as data for the seven experi-
ments were provided, the comparison for the remaining five cases will also be
presented.

| $d$ (mm) | **61** | 80 | 100 | **120** | 140 | 149 | 189 |
|---|---|---|---|---|---|---|---|
| $x_g$ (mm) (measured) | 551 | 617 | 696 | 763 | 846 | 877 | 1017 |
| $x_g$ (mm) (theoretical) | 560 | 630 | 705 | 780 | 854 | 888 | 1037 |
| $\eta_0$ (mm) | 13.0 | 9.2 | 7.8 | 5.1 | 4.4 | 4.2 | 3.1 |
| $a_0$ (m/s) | 1.20 | 1.21 | 1.19 | 1.17 | 1.14 | 1.20 | 1.21 |
| $u_t$ (m/s) | 1.70 | 1.64 | 1.93 | 2.03 | 2.13 | 1.94 | 1.97 |
| $t_0$ (s) | 1.42 | 1.36 | 1.62 | 1.74 | 1.87 | 1.62 | 1.63 |
| $S_0$ (m) | 2.408 | 2.223 | 3.130 | 3.522 | 3.980 | 3.136 | 3.207 |

Table 4: Measured and curve-fitted slide and wave parameters for the 7 experiments performed
by Enet and Grilli (2007). Nomenclature: Measured characteristic amplitude $\eta_0$ (at $x = x_0$).
Slide kinematics parameters $a_0, u_t$ and $t_0$.

In Figure 7 the comparison of the Multilayer-HySEA model numerical re-
sults with measured data for the first case, $d = 61\,mm$, in the four gauges, is

| $g_1$ | $g_2$ | $g_3$ | $g_4$ |
|-------|-------|-------|-------|
| $(x_0,0)$ | (1469,350) | (1929,0) | (1929,500) |

Table 5: Wave gauge locations $(x, y)$ in mm, as shown in Figure 6.

presented. An excellent agreement can be observed between these time series. The comparisons for the second required case ($d = 120\,mm$) in the 3 gauges with data provided (gauge $g_3$ was not available) are shown in Figure 8. Good agreement can also be observed in this case. Finally, Figure 9 shows the comparison for the five remaining cases provided by Enet and Grilli. In all cases (for all submergences), a good agreement between simulated results and measured lab data can be observed.

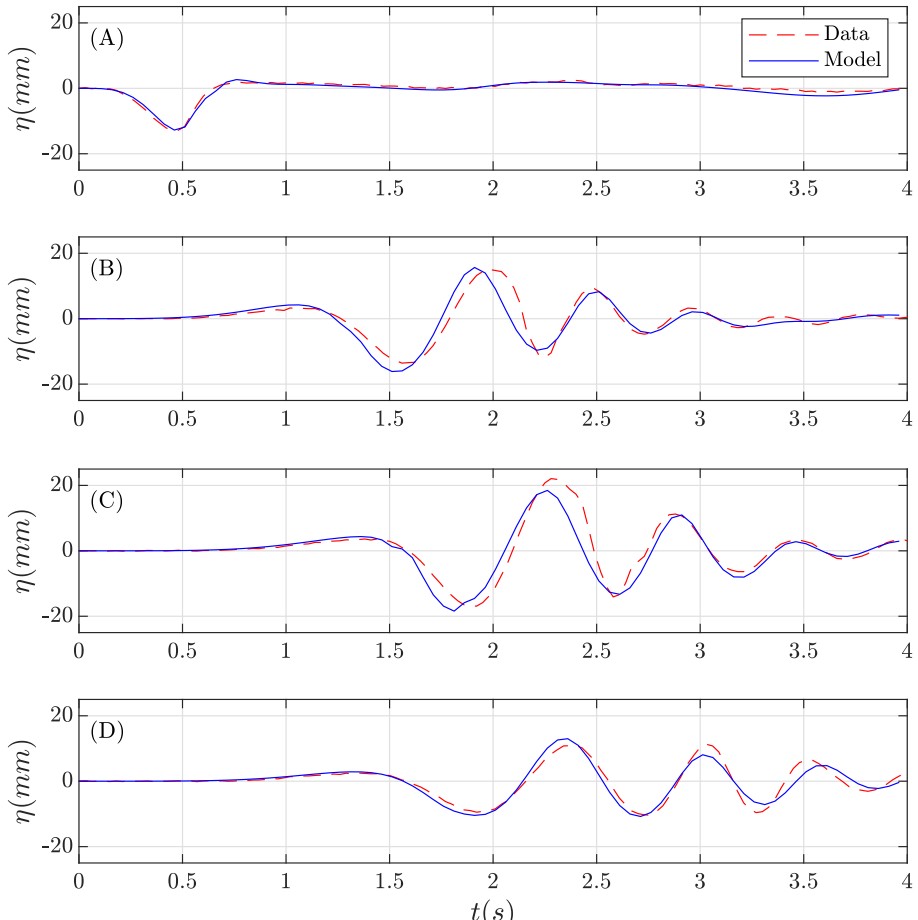

Figure 7: Test case $d = 61\ mm$. Numerically computed (in blue) time time series at wave gauges (A) $g_1$, (B) $g_2$, (C) $g_3$, and (D) $g_4$ compared with the lab measured data (in red).

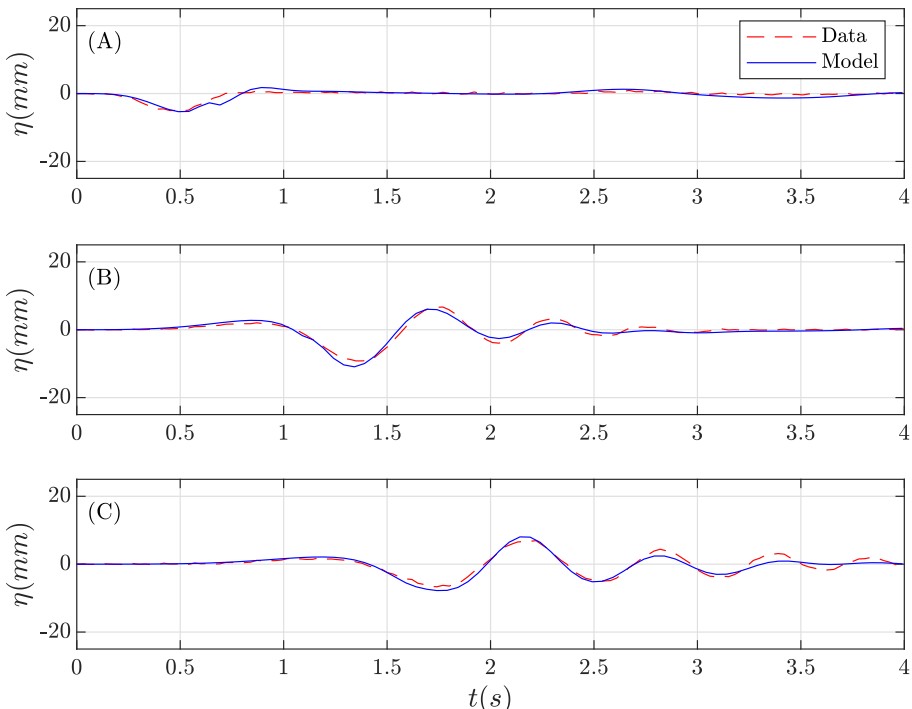

Figure 8: Test case $d = 120\ mm$. Numerically computed (in blue) time time series at wave gauges (A) $g_1$, (B) $g_2$, and (C) $g_4$ for compared with the lab measured data (in red).

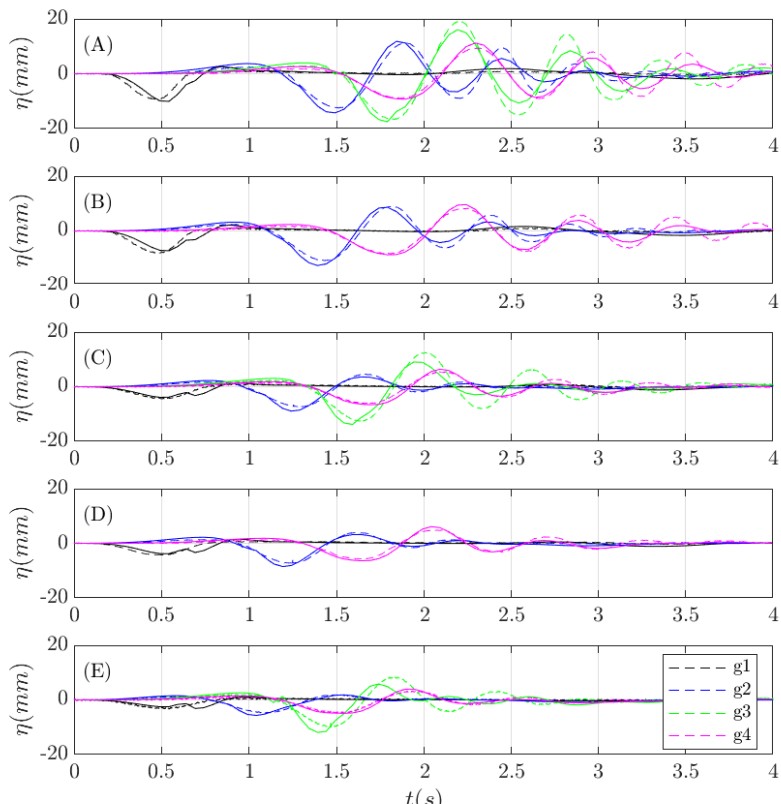

Figure 9: Comparison of data time series and numerical at wave gauges (dashed) for the cases (A) $d = 80\ mm$, (B) $d = 100\ mm$, (C) $d = 140\ mm$, (D) $d = 149\ mm$, and (E) $d = 189\ mm$.

In Table 6, the execution times for simulations performed on a NVIDIA Tesla P100 GPU are presented. It can be observed that including non-hydrostatic terms in the NLSW equations results in an increase of the computational time in 2.65 times. If a richer vertical structure is considered, then larger computational times are required. As examples for the two and three-layer systems, 3.3 and 4.45 times increase in the computational effort.

|       | Runtime (s) | Ratio |
|-------|-------------|-------|
| SWE   | 23.08       | 1     |
| 1L-NH | 61.20       | 2.65  |
| 2L-NH | 76.35       | 3.30  |
| 3L-NH | 102.93      | 4.45  |

Table 6: Execution times in seconds for SWE and non-hydrostatic GPU implementations. Ratios compared with SWE.

Figure 10 shows the comparison, for the four models considered, of the numerical results obtained with the measured data at gauge $g_4$ for the case $d = 189$ $mm$. It can be observed that a model vertical structure considering only one layer is not enough to reproduce the observed data, and that considering 2 and 3 layers in the model produce much better numerical results.

Moreover, Table 7 shows the period times, $T$, of the time series data in Figures 7, 8, and 9 for all the wave gauges. To obtain the vales for the periods, we have computed the elapsed time between the first two wave troughs in each time series. We have omitted the measurement for wave gauge $g_1$ because it was not clear how to measure the period in this case. Once the period from each time series has been measured, we have computed the wave number from the dispersion relation given from the Airy theory:

$$\frac{2\pi}{T} = \sqrt{gk \tanh(kH)}. \tag{23}$$

Table 8 shows the computed values $kH$ by solving the dispersion relation (23). On the view of the computed $kH$ values it can be stated that $kH \in [2.815, 4.528]$. Since multilayer models have good dispersion relation errors within this range

of $kH$ (see Table 1 and Figure 2), this explains the aforementioned excellent agreement between the computed time series and the measured lab data. Finally, although the phase velocity for the one-layer system shows an error bounded by only $3.02\,\%$ for $kH \in [0, 5]$ (see Table 1), it can be seen in Figure 10 that the one-layer non-hydrostatic pressure system cannot represent the waves correctly. In contrast, the one-layer system tends to amplify waves. This behaviour can be explained by observing the shoaling gradient for this model (see Figure 2). The shoaling gradient verifies the ODE

$$A' = -A \cdot \gamma(kH) \cdot \frac{\partial_x H}{H}, \tag{24}$$

where $A$ denotes the amplitude. Then, it can be stated by inspecting the solutions of the ODE (24) that if the shoaling gradient of the model $\gamma(kH)$ is underestimated with respect to the Airy theory ($\gamma < \gamma_{Airy}$), then the solutions of the system tend to amplify waves, in this case, for offshore wave propagation. The poor behavior shown by the one-layer system in some cases justifies the need to incorporate the improved multilayer model considered here.

| $d$ (mm) | 61 | 80 | 100 | 120 | 140 | 149 | 189 |
|---|---|---|---|---|---|---|---|
| $g_2$ | 0.69 | 0.686 | 0.704 | 0.69 | 0.676 | 0.692 | 0.8 |
| $g_3$ | 0.66 | 0.716 | – | – | 0.702 | – | 0.694 |
| $g_4$ | 0.84 | 0.8 | 0.784 | 0.75 | 0.794 | 0.784 | 0.751 |

Table 7: Measured wave period $T$ ($s$) for test cases $d = 61$, 120 (Figures 7 and 8) and $d = 80$, 100, 140, 149, 189 (Figure 9).

### 5.3. Benchmark Problem 3: Three-dimensional submarine/subaerial triangular solid block

This benchmark problem is based on the 3D laboratory experiment of Wu (2004) and Liu et al. (2005), for a series of triangular blocks of several aspect ratios moving down a plane slope into the water from a dry (subaerial) or wet (submarine) location. Figure 11 shows the schematic description of the set-up for this benchmark in the case of a partially submerged block. Further details

| $d$ (mm) | 61 | 80 | 100 | 120 | 140 | 149 | 189 |
|---|---|---|---|---|---|---|---|
| $g_2$ | 3.114 | 3.15 | 2.995 | 3.114 | 3.242 | 3.097 | 2.35 |
| $g_3$ | 4.528 | 3.85 | – | – | 4.004 | – | 4.097 |
| $g_4$ | 2.815 | 3.093 | 3.218 | 3.512 | 3.14 | 3.218 | 3.512 |

Table 8: Computed $kH$ values from the measured wave period (see Table 7) and the Airy dispersion relation $2\pi/T = \sqrt{gk\tanh(kH)}$ for test cases $d = 61,\ 80,\ 100,\ 120,\ 140,\ 149$ and 189.

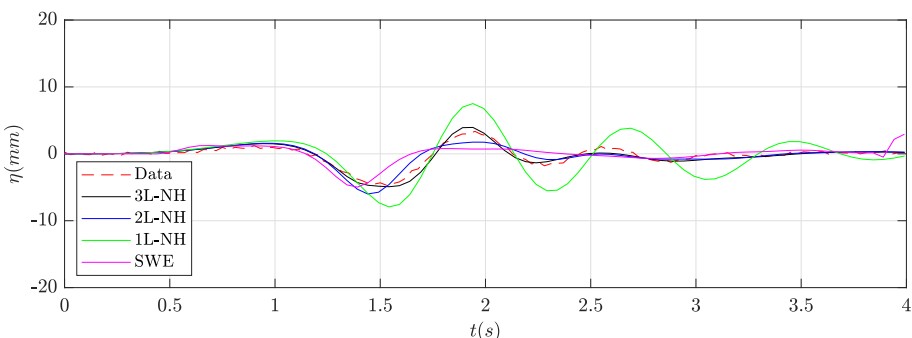

Figure 10: Test case $d = 189$ $mm$. Lab measured data (red) and numerically computed time series at wave gauge $g_4$ using different numerical models.

can be found at Kirby et al. (2018) and at LTMBW (2017). The laboratory experiments were conducted in a wave tank at Oregon State University of length 104 m, width 3.7 m, and depth 4.6 m.

A plane slope 1:2 (as the one shown in Figure 11 upper panel) with $\theta = 26.6°$ was located near one end of the tank and a dissipating beach in the other. In all the experiments, the water depth was $h_0 = 2.44\,\text{m}$. The experiments retained for the present benchmark were all performed with a triangular block of length $b = 0.91\,\text{m}$, width $w = 0.61\,\text{m}$, and vertical front face $a = b/2 = 0.455\,\text{m}$.

The block movement was provided by means of a polynomial fitting to measured data, giving the horizontal distance as:

$$x_{0,t} = x_{(0,t=0)} + (a\,t^3 + b\,t^2 + c\,t)\cos\beta, \tag{25}$$

with $\beta = \arctan(1/2)$ and $x_{(0,t=0)} = -2\Delta$. The polynomial coefficients for the two cases proposed are given in Table 9.

| $\Delta$ | $a$ | $b$ | $c$ |
|---|---|---|---|
| 0.10 m | -0.097588 | 0.759361 | 0.078776 |
| -0.25 m | -0.085808 | 0.734798 | -0.034346 |

Table 9: Polynomial coefficients defining slide motion.

For each case, measured free surface elevations for two wave gauges placed at $(x, y) = (1.83, 0)$ (in m) and $(x, y) = (1.2446, 0.635)$, where $x$ is the distance to the initial coastline and $y$ is the distance to the central cross-section (see location at Fig. 11 lower panel). Also measured runup for each case is given at runup gauges 2 and 3 in Figure 11 lower panel, lying on the slope at a distance $0.305\,\mathrm{m}$ and $0.611\,\mathrm{m}$ from the central cross-section, respectively.

The two-dimensional computational domain $[-2, 6] \times [-2, 2]$ is discretised with $\Delta x = \Delta y = 0.025\ m$ and the number of layers was set up to 3. Numerical experiments using more number of layers were performed, obtaining similar results. The stability CFL number was set to 0.9 and $g = 9.81$. The simulated time was 4 $s$. The same boundary conditions, as in the previous case, were imposed.

The numerical results obtained for the subaerial test case are presented in Figures 12 and 13. Figure 12 depicts the comparison for the time series at the wave gauges and Figure 13 at the runup gauges. The same comparison has been performed for the submerged test case, and it is presented in Figures 14 and 15. The agreement for the wave gauges is quite good for WG1 in both cases. For WG2, just in front of the block, an overshoot after the first depression wave is observed in both cases related to the turbulent nature of the experiment. Note that although a turbulent model is not considered here, we have noted that the breaking criteria helps to dissipate energy associated to this turbulent process. For the run-up, the qualitative agreement is quite good, with the larger discrepancies in RG3 for the submarine test case.

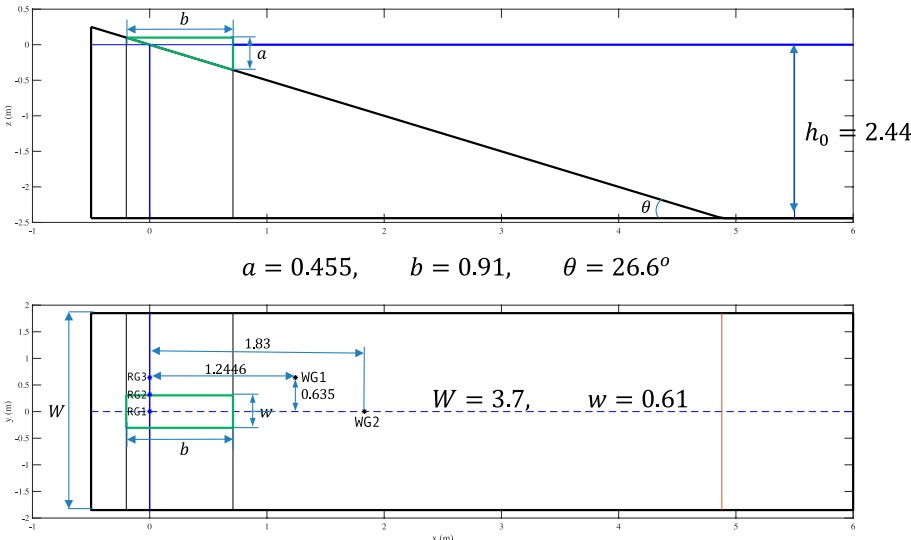

$$a = 0.455, \qquad b = 0.91, \qquad \theta = 26.6^o$$

$$W = 3.7, \qquad w = 0.61$$

Figure 11: Definition sketch for BP3 laboratory experiments. Here for a submerged ($\Delta < 0$) slide. Upper panel vertical cross section, lower panel plan view. All units are in meters.

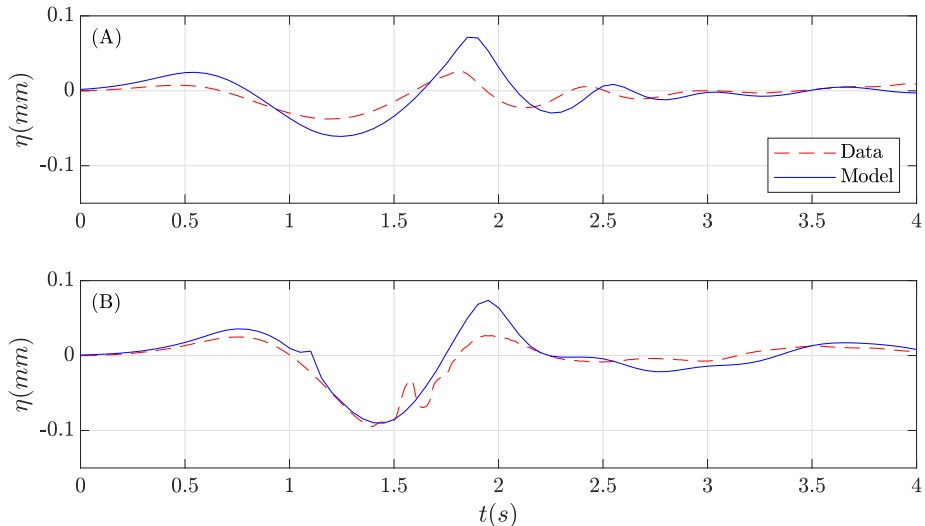

Figure 12: Subaerial test case. Lab measured water height (red) and numerical time series (blue) at wave gauges (A) WG1 and (B) WG2 .

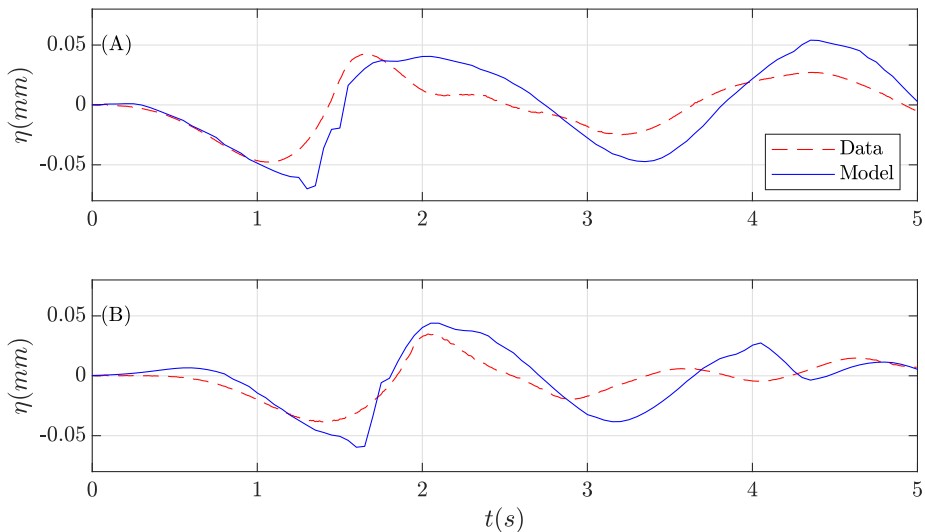

Figure 13: Subaerial test case. Lab measured runup (red) and numerical time series (blue) at runup gauges (A) RG2 and (B) RG3.

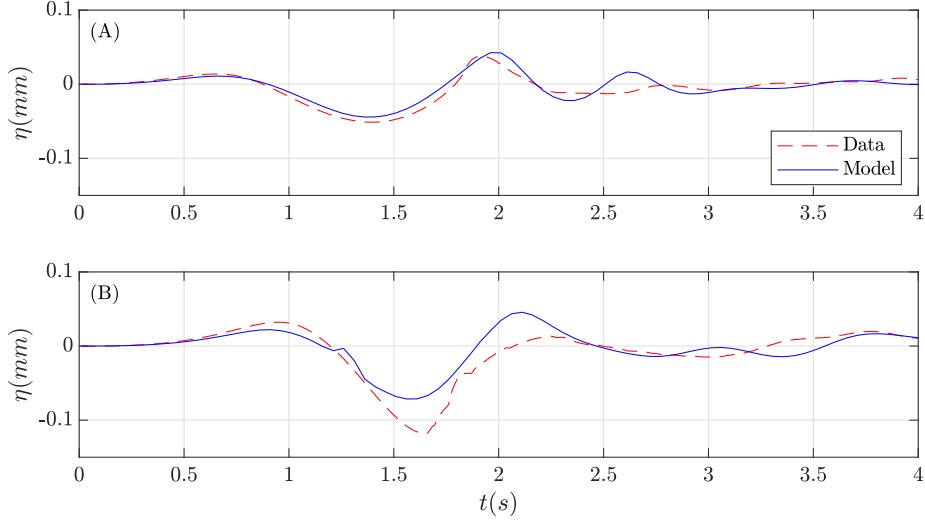

Figure 14: Submerged test case. Lab measured water height (red) and numerical time series (blue) at wave gauges (A) WG1 and (B) WG2.

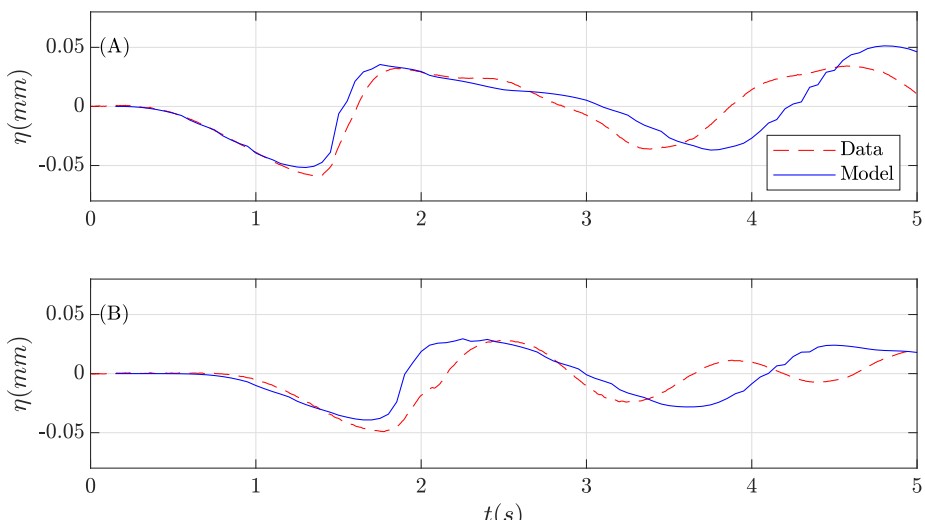

Figure 15: Submerged test case. Lab measured runup (red) and numerical time series (blue) at runup gauges (A) RG2 and (B) RG3.

## 6. Concluding Remarks

Validation of numerical models is a first unavoidable step before their use as predictive tools. This requirement is even more necessary when the developed models are going to be used for risk assessment in natural events where human lives are involved. The present work is the first step in this task for the Multilayer-HySEA model, a novel dispersive multilayer model of the HySEA suite developed at the University of Malaga. This model considers a stratified vertical structure and includes non-hydrostatic terms, this is done in order to include the dispersive effects in the propagation of the waves in a homogeneous, inviscid, and incompressible fluid. The numerical scheme implemented, combines a highly robust and efficient finite volume path-conservative scheme for the underlying hyperbolic system and finite differences for the discretization of the non-hydrostatic terms. In order to increase numerical efficiency, the numerical model is implemented to run in GPU architectures. In particular in NVIDIA graphics cards and using CUDA language. In the case of the traditional SW non-dispersive model, this kind of implementations produces an extremely efficient and fast code (Macías et al., 2020d). Increasing the number of layers in SW models provides an enhanced vertical resolution and, at the same time, increases the computational cost. Despite this, from a computational point of view, the two-layer non-hydrostatic code presents a good computational efficiency, and computing times with respect to the one layer SWE GPU code are absolutely reasonable, being only from 2 to 2.5 larger that for the one layer case. In the numerical simulations performed in the present work, the non-hydrostatic wall-clock times are always below 4.45 times those for the traditional SWE HySEA model, for a number of vertical layers up to three. The numerical scheme presented here and the corresponding multilayer SW water model proposed, is highly efficient and is able to model dispersive effects with a low computational cost.

Regarding model results, they show a good agreement with the experimental data for the three benchmark problems studied in the present work. In partic-

ular, for BP2, but this also occurs for the other two benchmark problems, we have shown that a one layer, hydrostatic or non-hydrostatic, model is not able to reproduce the complexity in the observed lab data considered in the proposed benchmarks. The waves to be modeled in the test cases proposed here are high-frequency and dispersive. Hence, it is at least necessary a two-layer structure and non-hydrostatic terms in the model to be used in order to capture the dynamics of the generated waves. As noted in Kirby et al. (2018) and in view of the results presented, the non-hydrostatic multilayer model discussed here can adequately represent the physics and behavior of the waves generated with a reasonable low computational cost.

## 7. Code and data availability

The numerical code used to perform the numerical simulations in this paper is available at HySEA codes web page at https://edanya.uma.es/hysea/index.php/download.

All the data used in the present work and necessary to reproduce the experiments set-up of the numerical experiments and the laboratory measured data to compared with, can be downloaded from LTMBW (2017) at the web site http://www1.udel.edu/kirby/landslide/. Finally, the NetCDF files containing the numerical results obtained with the Multilayer-HySEA code can be found and download from Macías et al. (2020b).

## 8. Authors' contributions

JM is leading the HySEA codes benchmarking effort undertaken by the EDANYA group, he wrote most of the paper, reviewed and edited it, assisted in the numerical experiments and in their set up. CE implemented the numerical code and performed all the numerical experiments, he also contributed to writing the manuscript. JM and CE did all the figures. MC strongly contributed to the design and implementation of the numerical code.

## 9. Competing interest

The authors declare that they have no conflict of interest.

## 10. Acknowledgements

This research has been partially supported by the Spanish Government-FEDER funded project MEGAFLOW (RTI2018-096064-B-C21), the Junta de Andalucía-FEDER funded project UMA18-Federja-161 and the University of Málaga, Campus de Excelencia Internacional Andalucía Tech.

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
