# Peer review of "Multilayer-HySEA model validation for landslide generated tsunamis. Part I Rigid slides"

_Natural Hazards and Earth System Sciences, 2020_

## Referee Comment (RC1) · Anonymous Referee #1 · 3 Sep 2020

General Comments:

This is an informative paper on state of the art tsunami modeling for land-slide generated tsunamis. The study presents results of performing simulations for 3 benchmark problems provided by the National Tsunami Hazard Mitigation Program (NTHMP) with the numerical code Landslide-HySEA. The topic and results presented in the paper are within the scope NHESSD topics. The paper provides a sufficiently detailed description of both, the governing equations modeled in the code and the numerical algorithm implemented to resolve the system. Additional references are provided for readers interested in additional details. The authors provide and adequate literature review of pre-existing validation efforts in the introductory section of the report and with a few necessary improvements also provide a clear and understandable description of the

numerical implementation of the laboratory experiments used for the validation. The results are presented in a clear and concise manner.

The first part of the paper could benefit from a thorough English language and stylistic review, particularly (Sections 1, 2). The style seems to improve after those two sections. It should also be reviewed for some typos:

Some examples typos are provided in the Specific Comments below.

Specific Comments:

pp1. Please correct the sentence: " The US National Tsunami has proposed the experimental data used Hazard and Mitigation Program (NTHMP) and established for the NTHMP Landslide Benchmark Workshop, held in January 2017 at Galveston." to " The US National Tsunami Hazard and Mitigation Program (NTHMP). . ."

pp2. l. 9; Please, provide reference for "Catalina Island " 2006 workshop.

pp3. l. 33; Fifteen or twenty?, please specify.

pp.4. Please, clarify the type of approximation the Landslide-HySEA uses to model the physical system. Are vertical pressure and velocity gradients modeled linearly within each layer with matching values at the interphase between layers?. What is exactly meant in line 74 by " The multilayer model is able to take into account the full vertical structure" isn't this an overstatement, please clarify.

pp. 5 Please, label equations throughout the paper.

pp. 5 (Equation System 1). There seems to be a lack of symmetry in the discretization of the continuity equation. Is this meant to be one-sided discretization?. Please, check.

pp 6. l. 109; Linearization around what "lake"?

pp. 8 Table 1. Please, include expression for Phase Velocity from Airy linear theory for reference. Why is only Phase Velocity approximations shown? What about Group

Velocity and Shoaling Gradient?

pp. 8 l. 145-147, Please, check if expected values of kH for your numerical experiment fall within the range properly modeled in your approximations (5<kH<15 pp7-8, l.144...), the paper would benefit from a brief discussion on this topic.

pp. 11. l. 168-174. Please, from the explanation on how the solution proceeds, the elliptic operator for the continuity equation seems to be computed implicitly, the pressure terms calculated with the solution from the continuity equation and then the discharge field updated in time explicitly? If this is the case, a bit more detail explanation would be appreciated. Otherwise, please clarify how the solution proceeds.

pp. 14, Figure 5. From the curves, the motion of the block seems to stop abruptly. Is this correct?

pp. 14, 223. Please, specify where in the domain outflow conditions are imposed.

pp. 14 & 23. Please, clarify how a Smagorinsky turbulent model for the Reynolds stresses is used in this context. Do the field equations preserve the classic Navier-Stokes viscous stress tensor for the resolved scales and a Smagorinsky model is then used for the Reynolds stresses?. The text seems to confuse these two terms as if they are the same one, but both are present separately in the Reynolds averaged Navier-Stokes equations.

pp. 17 Table 4. Please, define all variables in the column 1 of Table 4.

pp 23. l. 313; Please specify if delta x=delta y.

pp.23. Figure 12, please specify units in geometry parameters.

pp.26 l. 335; "...homogeneous, inviscid, and ..." is it inviscid, or have viscous terms been included as stated in pages 14 and 23? Please. correct.

pp.26 l. 355-356; The last sentence should be supported with results from full N-S models, otherwise I suggest to eliminate.

A couple of examples of typos that need correction:

pp. 1, Abstract; "The US National Tsunami has proposed the experimental data used Hazard and Mitigation Program (NTHMP) and established for the NTHMP Landslide Benchmark Workshop, held in January 2017 at Galveston."

pp. 2 l. 29; "...Multixlayer-HySEA..."

pp.10 l. 163; "...non-conservative hyperbolic system underlying system..."

pp. 21 l. 278; "Tesla P100 GPU In can be.."
* * *

---

## Referee Comment (RC2) · Anonymous Referee #1 · 9 Oct 2020

The authors seem to have intended to include a reference in their response that seems to be missing.

Please, correct "See ()", "given in ()",... in the response wherever they appear.

---

## Author Comment (AC2) · 9 Oct 2020

Reply to Anonymous Referee #1 - The problem is with NHESS LaTeX compilation

As the problem comes from the LaTeX compilation of the NHESS system when references are provided, (to make it simple) I include my compile pdf version as supplement below. I hope this it is ok with you.

Please also note the supplement to this comment:
https://nhess.copernicus.org/preprints/nhess-2020-171/nhess-2020-171-AC2-supplement.pdf
* * *
[Figure]

2020-171, 2020.

---

## Referee Comment (RC4) · Anonymous Referee #2 · 7 Dec 2020

General Comments:

The paper presents the state of the art description for numeric methods of tsunamis generated by land-slides. It presents a large number of references, which permits to apprehend the context of this work and what it brings in terms of novelty. The text clearly written which helps and readers follow the explanations, namely the results about the benchmarks. The subject and work is in the scope for publication in the NHESS journal.

Specific Comments

The figures displaying the comparison between the numeric modeling and laboratory data are clear and seem to indicate that the numeric code behaves well in reproducing

the time series registers at the laboratory acquired wave gauges. It maybe be a little exaggeration to say in pag 15 l. 284 "In Figure 6 the comparison of the numerical results with the filtered lab measured data is presented. An excellent overall agreement between them can be observed". In fact for figures 6C and 6D after ∼1 sec the two curves start to show an phase offset. Other figures like for example Figure 8 show a much better agreement.

I have, however, a more fundamental critique about one aspect of this work. It is said at pag 28, point 7 that addresses the "Code availability" topic that: "The numeric code is currently under development and only available to close collaborators". First, the code seems to be in sufficiently advanced state, which is what allowed to achieve the good comparison against the laboratory data shown in the benchmarks. But the part that says the code is closed source is what I think is more problematic for the intention to publish a paper about it. There is nothing wrong about people wanting to keep the codes they write in closed form. But in current times I think it is inappropriate to want both to publish a paper about the behavior of a numerical code and keep source code in closed form. It is thus my recommendation that publishing the paper, which has the quality and is in sate to be published, needs to fulfill the condition that the source code is made public for usage and review.

---

## Author Comment (AC3) · 9 Dec 2020

Regarding Figure 6 description in the text (now is Figure 5) we agree with the reviewer comment, and we have changed "excellent" by "good" agreement.

About the more fundamental observation regarding "Code availability" we will provide the version of the code used to perform all the benchmarks presented in the present study in our web page (as we already do with the open-source of Tsunami-HySEA) or in a repository. We will include the corresponding text in the section "Code availability".

---

## Author Response (AR1)

**General Comments:**

This is an informative paper on state of the art tsunami modeling for land-slide generated tsunamis. The study presents results of performing simulations for 3 benchmark problems provided by the National Tsunami Hazard Mitigation Program (NTHMP) withthe numerical code Landslide-HySEA. The topic and results presented in the paper are within the scope NHESSD topics. The paper provides a sufficiently detailed description of both, the governing equations modeled in the code and the numerical algorithm implemented to resolve the system. Additional references are provided for readers interested in additional details. The authors provide an adequate literature review of pre-existing validation efforts in the introductory section of the report and with a few necessary improvements also provide a clear and understandable description of the implementation of the laboratory experiments used for the validation. The results are presented in a clear and concise manner. The first part of the paper could benefit from a thorough English language and stylistic review, particularly (Sections 1, 2). The style seems to improve after those two sections. It should also be reviewed for some typos: Some examples typos are provided in the Specific Comments below.

*Authors comment: As suggested by the reviewer, we have sent the abstract and the first two sections for professional English language revision. This revised text has be included in the current version of the paper, now uploaded (This corrections are not marked in red in the revised verion).*

*First of all, note that the reviewer must have used for the revision an older version of the submitted paper as most of the typos he/she is mentioning are not present in the version that was uploaded in the NHESS system. Also, the numbering he/she is using does not correspond to that of the (up to today) current version of the paper.*

**Specific comments:**

- pp1 Please correct the sentence: "The US National Tsunami has proposed the experimental data used Hazard and Mitigation Program (NTHMP) and established for the NTHMP Landslide Benchmark Workshop, held in January 2017 at Galveston." to " The US National Tsunami Hazard and Mitigation Program (NTHMP)..."

  *This sentence was not present in the version uploaded at the NHESS system, corresponds to an older version.*

- pp2. l. 9; Please, provide reference for "Catalina Island" 2006 workshop.

  *The reference of Liu et al (2008) [7] has been included.*

- pp3. l. 33; Fifteen or twenty?, please specify.

  *Again, this is not in the version we are taking from NHESS system. In that version, it can be read "Twenty years ago, at the beginning of the century,..."*

- pp.4. Please, clarify the type of approximation the Landslide-HySEA uses to model the physical system. Are vertical pressure and velocity gradients modeled linearly within each layer with matching values at the interphase between layers?

  *The system is derived under the assumption of a linear vertical profile within each layer for the total pressure. The horizontal and vertical velocities are assumed to have a constant vertical profile within each layer. More details on the derivation of the system (1) can be found in [5] (model $LDNH_0$).*
  *An alternative deduction of the same system (1) is performed in [3] assuming linear vertical profiles for pressure and vertical velocity, and constant vertical profile for the horizontal velocity, as well as some extra hypothesis for the case of two layers.*
  *Concerning the continuity of variables at the interphase between layers:*
  *The system is derived from Euler equations after a Galerkin approach at z-direction. At the discrete level on z, the total pressure matches at interphases and velocities satisfies a discrete jump condition (See [5] or [3]).*

  This clarifications have been added in the first two paragraphs of Section 3.

- What is exactly meant in line 74 by "The multilayer model is able to take into account the full vertical structure" isn't this an overstatement, please clarify.

  *We wanted to express that the multi-layer model is able to better approximate the vertical structure than a standard one-layer depth-averaged model. Considering a richer vertical structure provides better approximations of more complex 3D effects that may occur in the simulated flow. In particular, increasing the number of layers, the linear dispersion relation of the model converges towards the exact dispersion relation from the Stokes linear theory (see [5] for a proof of the result).*

  Three sentences, including the text above, have been added at the end of the paragraph in order to clarify this point.

- pp. 5 Please, label equations throughout the paper.

  *Done.*

- pp. 5 (Equation System 1). There seems to be a lack of symmetry in the discretization of the continuity equation. Is this meant to be one-sided discretization? Please, check.

  *The system (1) is equivalent to the model $LDNH_0$ given in [5] for a given number of layers L.*

  *The system $LDNH_0$ contains exactly L equations corresponding to the incompressibility condition at each layer (see eq. (3.18a) in [5]). However, given a layer $\alpha \in \{1, \ldots, L\}$, the incompressibility eq. (3.18a) involves the presence of the spatial derivatives of $u_1$, $u_2$, $\ldots, u_{\alpha-1}$, as well as $u_\alpha$, that makes the design of the numerical method less efficient.*

  *In the present paper, we have circumvented that difficulty by doing some algebraical manipulations.*
  *Let us show the equivalence between incompressibility equations given in (1), and the incompressibility equations of system $LDNH_0$, given in eq. (3.18a) [5].*

- *Procedure to obtain the incompressibility equation for the lower layer $\alpha = 1$ in (1):*
  *Equation (3.18a) given in [5] for the lower layer $\alpha = 1$ and written with the notation used in this reviewed paper reads:*

$$w_1 + \partial_t H - u_1 \partial_x \left( -H + \Delta s h \right) + \frac{1}{2} \partial_x \left( \Delta s h u_1 \right).$$

  *Now, if we divide by $\Delta s h > 0$ at both sides of the previous equation and after arranging terms, we obtain:*

$$\frac{1}{2} \partial_x u_1 + \frac{1}{h \Delta s} u_1 \partial_x \left( H - \frac{1}{2} h \Delta s \right) + \frac{1}{h \Delta s} \left( w_1 + \partial_t H \right) = 0,$$

  *that coincides with the incompressibility equation given in (1) for the lower layer $\alpha = 1$:*

$$\partial_x u_{1/2} + \sigma_{1/2} \partial_z u_{1/2} + \partial_z w_{1/2} = 0,$$

  *where, $u_{1/2}$, $\sigma_{1/2}$, $\partial_z u_{1/2}$ and $\partial_z w_{1/2}$ are given in eqs. (2), (3) and (5):*

$$u_{1/2} = \frac{u_1 + u_0}{2} = \frac{u_1 + 0}{2}, \quad \sigma_{1/2} = \partial_x \left( H - \frac{1}{2} h \Delta s \right),$$

$$\partial_z u_{1/2} = \frac{1}{h \Delta s} \left( u_1 - u_0 \right) = \frac{1}{h \Delta s} \left( u_1 - 0 \right),$$

$$\partial_z w_{1/2} = \frac{1}{h \Delta s} \left( w_1 - w_0 \right) = \frac{1}{h \Delta s} \left( w_1 + \partial_t H \right).$$

- *Procedure to obtain the incompressibility equation for a given layer $\alpha > 1$ in (1):*
  *Equation (3.18a) given in [5] for the layer $\alpha$ and $\alpha - 1$, and written with the notation used in this reviewed paper, respectively reads:*

$$w_\alpha + \partial_t H - u_\alpha \partial_x \left( -H + (\alpha - 1/2) \Delta s h \right) + \partial_x (\Delta s h u_1) + \ldots + \partial_x (\Delta s h u_{\alpha-1}) + \frac{1}{2} \partial_x (\Delta s h u_\alpha) = 0,$$

  *and*

$$w_{\alpha-1} + \partial_t H - u_{\alpha-1} \partial_x \left( -H + (\alpha - 1 - 1/2) \Delta s h \right) + \partial_x (\Delta s h u_1) + \ldots + \partial_x (\Delta s h u_{\alpha-2}) + \frac{1}{2} \partial_x (\Delta s h u_{\alpha-1}) = 0.$$

  *Now, subtracting both equations yields*

$$\Delta s h \partial_x \left( \frac{u_\alpha + u_{\alpha-1}}{2} \right) + w_\alpha - w_{\alpha-1} + \left( u_\alpha - u_{\alpha-1} \right) \partial_x \left( H - (\alpha - 1) \Delta s h \right) = 0.$$

  *If we divide at both sides of the previous equation by $\Delta s h > 0$, then we can re-write it using the notation introduced in the present reviewed paper:*

$$\partial_x u_{\alpha-1/2} + \sigma_{\alpha-1/2} \partial_z u_{\alpha-1/2} + \partial_z w_{\alpha-1/2} = 0$$

  *that coincides with the expression given in (1).*

*Therefore, if we consider the matrix operator associated with the set of incompressibility equations, then it is clear that this operator (from eq (3.18a) in [5], consists of a lower triangular matrix.*
*However, with the procedure followed here, we can pre-compute a triangularization of the original operator, that can be solved later in a more efficient way.*

The above explanation has not been included in the corrected version of the paper now uploaded, nevertheless we could include it as a Remark or Appendix if the reviewer considers it is appropriate.

- pp 6. l. 109; Linearization around what "lake"?

  *The word "lake" does not appear in the current version. It was in an older version.*

- pp. 8 Table 1. Please, include expression for Phase Velocity from Airy linear theory for reference. Why is only Phase Velocity approximations shown? What about Group Velocity and Shoaling Gradient?

  *We have included the expression for Phase Velocity from Airy linear theory as requested (see eq (7)).*
  *Concerning Group Velocity and Shoaling Gradient, we have considered that the expressions are too tedious to be given here. Instead, we present the relative errors of phase and group velocities as well as linear shoaling with respect to the Airy theory in Figure 2.*

- pp. 8 l. 145-147, Please, check if expected values of $kH$ for your numerical experiment fall within the range properly modeled in your approximations ($5 < kH < 15$ pp7-8, l.144...), the paper would benefit from a brief discussion on this topic.

  *We agree with the reviewer's comment. We have included a discussion on this topic for the benchmark problem 2, where it was clear how to measure the experiment's wave number.*

- pp. 11. l. 168-174. Please, from the explanation on how the solution proceeds, the elliptic operator for the continuity equation seems to be computed implicitly, the pressure terms calculated with the solution from the continuity equation and then the discharge field updated in time explicitly? If this is the case, a bit more detail explanation would be appreciated. Otherwise, please clarify how the solution proceeds.

  *We have detailed the explanation in the present reviewed paper.*
  *The procedure is based on the Chorin's projection method, a standard and well-known approximation for dispersive systems (see, for instance, [1, 3, 4, 6, 9, 11].*

- pp. 14, Figure 5. From the curves, the motion of the block seems to stop abruptly. Is this correct?

  *Is correct. We use the prescribed motion for the block given in eq (17). Moreover, the block stops at time $t = t_0 = 1.677$ s. This is part of the benchmark description provided by the NTHMP. For the workshop we presented results with a non-stopping slide but we were simulating a different experiment. This information has been included in the reviewed paper.*

- pp. 14, 223. Please, specify where in the domain outflow conditions are imposed.

  *Done. In the case of this 2D experiment, outflow boundary conditions were imposed at $x = -1$, $x = 10$.*

- pp. 14 & 23. Please, clarify how a Smagorinsky turbulent model for the Reynolds stresses is used in this context. Do the field equations preserve the classic Navier-Stokes viscous stress tensor for the resolved scales and a Smagorinsky model is then used for the Reynolds stresses? The text seems to confuse these two terms as if they are the same one, but both are present separately in the Reynolds averaged Navier-Stokes equations.

  *We do not pretend to perform Navier-Stokes simulations. We understand that this is not clear in the present wording and a new subsection has been included to better explain this issue (see Subsection 3.2. Wave breaking and wetting and drying).*
  *It is well known that dispersive systems can not correctly represent breaking waves, and a breaking mechanism has to be included to account for thesebreaking effects. In the current*

*work, we add to the equation a dissipation term in momentum equations that accounts for the dissipation of the energy associate with a breaking bore. The key idea is that, in the presence of a breaking bore, a hydrostatic regime can better represent the physics.*
*To do that, we have included the diagonal part of the stress tensor for the horizontal momentum equation, and a breaking criteria is used to switch on and off such dissipation terms near to breaking bores. This is a standard technique when dealing with such type of dispersive problems and breaking waves (see [3, 4, 6, 11] and references therein.)*

This is detailed in the new Subsection 3.2 "Modeling of breaking waves and wetting and drying treatment".

- pp. 17 Table 4. Please, define all variables in the column 1 of Table 4.

  *Done.*
  *Some of the parameters appearing in Table 4 in the former version of this paper have been removed since they were expendable for the experiment's description. However, the interested reader is referred to the original paper [2].*

- pp 23. l. 313; Please specify if delta x=delta y.

  *Done.*

- pp.23. Figure 12, please specify units in geometry parameters.

  *Done in the caption. It also appears in the axis labels.*

- pp.26 l. 335; "...homogeneous, inviscid, and..." is it inviscid, or have viscous terms been included as stated in pages 14 and 23? Please. correct.

  *It is inviscid. The included "Viscous terms" consist of a standard technique to simulate the dissipation of energy associated with breaking waves that allow correct treatment of breaking bores.*

- pp.26 l. 355-356; The last sentence should be supported with results from full N-S models, otherwise I suggest to eliminate.

  *Done.*

- A couple of examples of typos that need correction: *No need to correct them, as they were not in the previous version of the paper*

  - pp. 1, Abstract; "The US National Tsunami has proposed the experimental data used Hazard and Mitigation Program (NTHMP) and established for the NTHMP Landslide Benchmark Workshop, held in January 2017 at Galveston."

    *Not present in the version at the NHESS system. Corresponds to an older version*

  - pp. 2 l. 29; "...Multixlayer-HySEA..."

    *Same thing. Not in the current version.*

  - pp.10 l. 163; "...non-conservative hyperbolic system underlying system..."

    *Not present in the current version.*

  - pp. 21 l. 278; "Tesla P100 GPU In can be.."

    *Not present in the current version.*

Anonymous Referee #2

Regarding Figure 6 description in the text (now is Figure 5) we agree with the reviewer comment, and we have changed "excellent" by "good" agreement.

About the more fundamental observation regarding "Code availability", the version of the code actually used to perform all the benchmarks presented in the present study can be downloaded from our web page (as we already do with the open-source of Tsunami-HySEA since 2016). The section "Code and data availability" now stands as follows:

"The numerical code used to perform the numerical simulations in this paper is available at HySEA codes web page at https://edanya.uma.es/hysea/index.php/download.

All the data used in the present work and necessary to reproduce the experiments set-up of the numerical experiments and the laboratory measured data to compared with, can be downloaded from [8] at the web site http://www1.udel.edu/kirby/landslide/. Finally, the NetCDF files containing the numerical results obtained with the Multilayer-HySEA code can be found and download from [10]."

**Bibliography**

[1] A.J. Chorin. Numerical solution of the navier-stokes equations. *Mathematics of Computation*, 22(104):745–762, 1968.

[2] F. Enet and S.T. Grilli. Experimental study of tsunami generation by three-dimensional rigid underwater landslides. *Journal of Waterway, Port, Coastal, and Ocean Engineering*, 133(6):442–454, 2007.

[3] C. Escalante, E.D. Fernández-Nieto, T. Morales, and M.J. Castro. An efficient two–layer non–hydrostatic approach for dispersive water waves. *Journal of Scientific Computing*, 2018.

[4] C. Escalante, T. Morales, and M.J. Castro. Non-hydrostatic pressure shallow flows: GPU implementation using finite volume and finite difference scheme. *Applied Mathematics and Computation*, 338:631 – 659, 2018.

[5] E.D. Fernández, F. Bouchut, D. Bresh, M.J. Castro, and A. Mangeney. A new Savage-Hutter type model for submarine avalanches and generated tsunami. *J. Comp. Phys.*, 227:7720–7754, 2008.

[6] M. Kazolea and A. I. Delis. A well-balanced shock-capturing hybrid finite volume-finite difference numerical scheme for extended 1D Boussinesq models. *Applied Numerical Mathematics*, 67(1):167–186, 2013.

[7] P. L-F Liu, H. Yeh, and C. Synolakis. *Advanced Numerical Models for Simulating Tsunami Waves and Runup*. WORLD SCIENTIFIC, 2008.

[8] LTMBW. Landslide Tsunami Model Benchmarking Workshop, Galveston,Texas, 2017. http://www1.udel.edu/kirby/landslide/index.html, 2017. Accessed: 2020-04-25.

[9] G. Ma, F. Shi, and J.T. Kirby. Shock-capturing non-hydrostatic model for fully dispersive surface wave processes. *Ocean Modelling*, 43–44:22–35, 2012.

[10] J. Macías, C. Escalante, and M.J. Castro. Numerical results in Multilayer-HySEA model validation for landslide generated tsunamis. Part I Rigid slides. Dataset on Mendeley, 2020. doi: 10.17632/xtfzrbvcb2.1.

[11] M. Ricchiuto and A. G. Filippini. Upwind residual discretization of enhanced Boussinesq equations for wave propagation over complex bathymetries. *Journal of Computational Physics*, 271:306–341, 2014.